# On Non-Completeness and G-Equivariance

**Yoo-Jin Baek**

Department of Information Security, Woosuk University, Jeollabuk-do 55338, Korea; yoojin.baek@gmail.com;
Tel.: +82-63-290-1221

**Abstract:** With the growing threat of the side-channel attack (SCA) to the cryptographic algorithm's implementations, the masking method has become one of the most promising SCA countermeasures for securely implementing, for example, block ciphers. The basic principle of the masking method is that if the sensitive variable (which, by definition, depends on sensitive information) is split into some random variables and they are manipulated in a secure manner, then the relationship between the random variables and the corresponding side-channel information may look independent from the outside world. However, after the introduction of the glitch attack, there has been a lot of concern about the security of the masking method itself. And, to mitigate the threat of the glitch attack, the threshold implementation (TI) and G-equivariant gates were independently introduced as countermeasures. In this paper, we consider the main notions of two such independent glitch attack's countermeasures, say, non-completeness and G-equivariance, and investigate their relationship. The contribution of this paper is three-fold. First, we show that the widely-circulated proof that the non-complete TI with uniform inputs guarantees the security against the 1st order DPA even in the presence of glitches is not satisfactory. Next, using the extended notion of G-equivariance to the higher-order setting, we prove that non-completeness implies G-equivariance, which, in turn, means that the non-complete TI with uniform inputs has resistance against the glitch attack. Thirdly, we prove that the set of non-complete gates is a proper subset of the set of G-equivariant gates by showing there is a gate that is G-equivariant but not non-complete.

**Keywords:** threshold implementation; G-equivariance; side-channel attack; masking method; glitch attack

## 1. Introduction

With the growing threat of SCA (Side-Channel Attack, [1–4]), many countermeasures have been proposed accordingly, and the masking method has been one of the most promising power attack countermeasures for securely implementing block ciphers [5,6].

The basic principle of the masking methods is that if the sensitive variable (which depends on key material and some known information by definition) is split into some random shares and then is manipulated in a secure manner, the relationship between the behavior of internal variables and the corresponding side-channel information may look independent from the outside world. However, after the introduction of the glitch attack, there has been a lot of concern about the security of the masking method itself [7–9].

The glitch attack primarily utilizes a specific hardware (HW) criterion, say, the glitch. More precisely, most of the HW-based masking schemes prior to the glitch attack were mainly based on the assumption that all gates' input signals arrive at the gate simultaneously. However, that is not true in practice. That is, the gate delay and the variable path lengths are very common in semiconductor technologies, so each input signal arrives at a gate at different times. And, with this phenomenon in hand, the state of the gate's output signal fluctuates within a clock cycle until it is

finally stabilized at a certain value, which is called a glitch. In particular, the glitch phenomenon highly affects the amount of electrical power consumed by the circuit, and, moreover, the consumed power is highly related to the circuit's input. The glitch attack analyzes this relationship to retrieve some secret information in cryptographic devices. Unfortunately, most masking schemes prior to the glitch attack have been shown to possess some inherent vulnerabilities to the attack.

To mitigate the devastating effect of the glitch attack, the authors in [10–13] introduced the concept of TI (Threshold Implementation), which is based on the techniques of secret-sharing and multi-party computation and is believed to be provably secure against a first-order differential power attack (DPA). Basically, TI is a Boolean masking scheme which satisfies three specific properties, say, correctness, non-completeness, and uniformness. The correctness implies that the (Boolean) sum of the output shares is equal to the original output. And, the ($d$-th order) non-completeness means that processing any combination of up to $d$ output masks altogether does not require at least one input share. The uniformness requires that the output distribution is preserved when a function is implemented in a shared form. The exact definition of these three properties is given in the next section. The most important characteristic of TI is that (first-order) TI is provably secure against the 1st-order differential power attack even in the presence of glitches [11].

As another glitch attack countermeasure, the authors in [14] proposed a new kind of gates, say, G-equivariant gates. Simply speaking, the G-equivariant gate is the gate, the toggling count of whose output is independent with the arrival order of the gate's input signals. After all, the independency is believed to successfully prevent the 1st-order side-channel DPA regardless of the arrival order of the input signals. However, it was also shown in [14] that there are no G-equivariant gates which have two input shares and the XOR sum in which two output shares is equal to the logical AND evaluation of its original inputs.

Contribution of this paper. As noted before, the most notable characteristic of TI is that it is provably secure against the 1st-order DPA even in the presence of glitches. And, the main concern of this paper lies at the security proof itself. That is, we re-investigate the proof that TI is secure against the 1st-order DPA even in the presence of glitches, and conclude that the proof has some missing points. To remedy it, we then propose to utilize the concept of G-equivariance. More precisely, while the original concept of G-equivariance is defined for 2-share masking setting, we use the extended notion of G-equivariance to an arbitrary number of shares and prove that the non-complete masked circuit is actually G-equivariant, which, in turn, implies that any non-complete TI can successfully prevent the glitch attack. Finally, we show that there are some G-equivariant gates that are not non-complete; thus, the notion of G-equivariance is broader than that of non-completeness. Consequently, the contribution of this paper is three-fold. First, we review the proof in [14] that any non-complete TI with uniform input sharing is secure against the 1st order DPA even in the presence of glitches and claim that the proof is not satisfactory by introducing a contradictory example. Next, extending the definition of G-equivariance, this paper proves that non-completeness implies G-equivariance, which fills the gap in the security proof in [14]. Finally, we give some exemplary gates which are not non-complete but are G-equivariant.

The paper is organized as follows. In Section 2, we review some prerequisite knowledge for our subsequent discussions. In Section 3, we show why the proof that non-completeness guarantees the 1st-order DPA-resistance is not satisfactory. Then, in the next section, we will prove that non-completeness implies G-equivariance and give some examples that are G-equivariant without being non-complete. The conclusion is given in Section 5.

## 2. Prerequisite

### 2.1. Notation

In this paper, small letters (possibly with superscripts) stand for elements of finite fields or functions over finite fields and small letters (possibly with superscripts) with subscripts are used

for denoting a component element of a finite field element (when the finite field is considered as a vector space over a base field), a component function of a vectorial function, or an input parameter of a multivariate function. For example, $a \in \mathrm{GF}(2^n)$ can be written as $a = (a_0, a_1, \ldots, a_{n-1})$ for $a_i \in \mathrm{GF}(2)$. The $n$-th order mask of $a \in \mathrm{GF}(2^n)$ (which will more precisely be defined later) is denoted as $(a^0, x^1, \ldots, a^{n-1})$ for $a^i \in \mathrm{GF}(2^n)$.

## 2.2. Masking Method

Power attack, which was first introduced by P. Kocher et al. can retrieve sensitive information in cryptographic devices using devices' power consumption patterns ([2]). Various techniques were introduced as power attack countermeasures, among which the masking method is representative for securely implementing, for example, block ciphers against DPA(differential power attack).

To apply the (Boolean) masking method to a function $z = f(x) : \mathrm{GF}(2^n) \to \mathrm{GF}(2^m)$, one should first determine the input masking order $d_{in}$ and the output masking order $d_{out}$ and then should take the following procedure: for given $x \in \mathrm{GF}(2^n)$,

(1) Randomly choose $x^1, \ldots, x^{d_{in}} \in \mathrm{GF}(2^n)$.

(2) Compute $x^0 = x \oplus x^1 \oplus \cdots \oplus x^{d_{in}}$, where $\oplus$ denotes the bitwise eXclusive-OR operation.

(3) From $(x^0, x^1, \ldots, x^{d_{in}})$, compute $(z^0, z^1, \ldots, z^{d_{out}}) \in \mathrm{GF}(2^m)^{d_{out}+1}$ with $z^0 \oplus z^1 \oplus \ldots \oplus z^{d_{out}} = f(x)$ in the manner that any information about the original input $x$ is not leaked during the computation.

The procedure above is usually called the $(d_{in}, d_{out})$ -order masking scheme for $z = f(x)$ and, if $d_{in} = d_{out} = d$, it is also called the $d$-th order masking scheme. The vector $(x^0, x^1, \ldots, x^{d_{in}})$ is called the $d_{in}$-th order mask (or, sharing) of $x$ or the $d_{in}$-th order input mask of $z = f(x)$ and each $x^i$ is called a share of $x$. Similarly, $(z^0, z^1, \ldots, z^{d_{out}})$ is called the $d_{out}$-th order mask of $z$ or the $d_{out}$-th order output mask of $z = f(x)$, and each $z^i$ is called a share of $z$. Importantly, the quantities $d_{in}$ and $d_{out}$ are closely related to the effort an attacker has to pay to break the masking scheme. For example, to successfully recover a key from the masking scheme with $d_{in}$-th input order, it is believed that an attacker needs to observe at least $d_{in} + 1$ individual shares or statistical moments.

The function computing the output mask $(z^0, z^1, \ldots, z^{d_{out}})$ given the input mask $(x^0, x^1, \ldots, x^{d_{in}})$ is denoted as $(f^0, \ldots, f^{d_{out}})$ and is called a shared implementation of $z = f(x)$, thus $f^i(x^0, x^1, \ldots, x^{d_{in}}) = z^i$ for each $i$.

Devising a masking scheme for linear or affine functions is known to be an easy task. For example, if $z = f(x)$ is linear (with respect to $\oplus$) and $x = x^0 \oplus x^1 \oplus \cdots \oplus x^{d_{in}}$, then $f(x) = f(x^0 \oplus x^1 \oplus \cdots \oplus x^{d_{in}}) = f(x^0) \oplus \cdots \oplus f(x^{d_{in}})$, thus letting $f^i(x^0, x^1, \ldots, x^{d_{in}}) = f(x^i)$ gives a well-established masking scheme for $f$. However, designing a masking scheme for non-linear functions, such as block ciphers' S-box, is non-trivial, and the gate-level masking method was introduced to address this issue, especially in the hardware masking scheme design [15].

The idea of the gate-level masking method is very simple: after decomposing any function (or circuit) into basic gates, like AND, XOR and so on, and individually applying an appropriate masking scheme to the corresponding basic gates, the resulting circuit will serve as the masking scheme for the original function. Especially, since AND, OR, NAND, and NOR gates are the only non-linear basic gates, and OR, NAND, and NOR gates can be constructed with the composition of AND, XOR, and NOT gates; the main research of the gate-level masking method focuses on how to apply the masking method to the AND gate. For example, Figure 1 shows the masking scheme for the 2-input AND gate proposed by E. Trichina [15], which can be mathematically described as: for a random bit $r$,

$$(x^0, x^1, y^0, y^1) \to (z^0, z^1) = ((((r \oplus x^0 y^0) \oplus x^1 y^0) \oplus x^0 y^1) \oplus x^1 y^1, r). \tag{1}$$

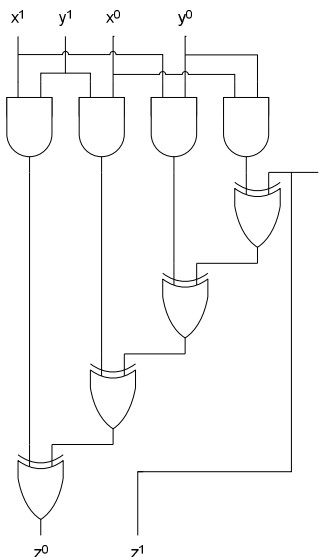

**Figure 1.** Masked AND gate by E. Trichina.

### 2.3. Glitch Attack and Countermeasures

Most of the previous gate level masking methods prior to the glitch attack implicitly or explicitly assumed that all input signals of any gate arrive at the gate simultaneously. However, the idealized assumption is shown not to hold in practice. That is, due to the gate delay and the variable path lengths, which are very common in the semiconductor technology, each input signal arrives at the gate at different times. Moreover, eliminating such the arrival time variation is known to be a hard task, especially for CMOS, the most widely used semiconductor technology. And, when input signals get to a gate at a different time, the state of the gate's output signal fluctuates within a clock cycle until it is finally stabilized at a certain value. This phenomenon is called the glitch or the hazard.

The glitch phenomenon highly affects the amount of electrical power consumed by circuits. Moreover, the amount of consumed power is highly related to the circuit's input. And, the power analysis attack which analyzes the relationship between the glitch phenomenon and the power consumption pattern is called the glitch attack [7–9]. Unfortunately, many gate-level masking schemes are known to be vulnerable to the glitch attack. Various countermeasures have been proposed since the glitch attack was introduced, and this paper focuses on the G-equivariant gates [14] and the threshold implementation [10–13].

The notion of G-equivariant gates relies on the belief that if a (averaged) toggling count of the gate's output is constant regardless of the arrival order of the gate input signals [14], the gate's power consumption pattern will not be influenced by the glitch phenomenon so that the glitch attack can be prevented. Unfortunately, as noted in [14], there are no 2-share G-equivariant gates in which the XOR sum gives rise to the AND-gate evaluation of the original input values. More precisely, there are no G-equivariant gates $g^0$, $g^1 : \mathrm{GF}(2)^4 \rightarrow \mathrm{GF}(2)$ satisfying

$$g^0(x^0, x^1, y^0, y^1) \oplus g^1(x^0, x^1, y^0, y^1) = (x^0 \oplus x^1)(y^0 \oplus y^1). \tag{2}$$

To remedy this undesirable situation, the authors introduced the concept of semi-G-equivariance with the weakened condition that, in the Equation (2), $x^0, x^1$ arrives at the gate at the same time and $y^0, y^1$ arrives at the gate simultaneously. However, even though there are some semi-G-equivariant gates that are available for glitch-attack-resistantly constructing the AND gate, there have been some doubts about the appropriateness of such weakened conditions.

TI (Threshold Implementation, [10–13]) was proposed as another glitch attack countermeasure and is said to be provably secure even in the presence of glitches. TI is defined to be a masking scheme

satisfying three specific properties: correctness, non-completeness, and uniformness. Given a function $z = f(x)$, the shared implementation $(f^0, \ldots, f^{d_{out}})$ of $f$ is said to be correct if

$$f^0(x^0, x^1, \ldots, x^{d_{in}}) \oplus \cdots \oplus f^{d_{out}}(x^0, x^1, \ldots, x^{d_{in}}) = f(x^0 \oplus x^1 \oplus \cdots \oplus x^{d_{in}})$$

for any mask $(x^0, x^1, \ldots, x^{d_{in}})$ of $x$. And $(f^0, \ldots, f^{d_{out}})$ is defined to be $d$-th order non-complete if any processing of up to $d$ component functions $f^i$ does not require at least one input share. Thus, for example, the following $(f^0, f^1, f^2)$ is the first-order non-complete shared implementation of the AND gate: for $x^0, x^1, x^2, y^0, y^1, y^2 \in GF(2)$,

$$f^0(x^0, x^1, x^2, y^0, y^1, y^2) = x^1 y^1 \oplus x^1 y^2 \oplus x^2 y^1$$

$$f^1(x^0, x^1, x^2, y^0, y^1, y^2) = x^2 y^2 \oplus x^2 y^0 \oplus x^0 y^2 \tag{3}$$

$$f^2(x^0, x^1, x^2, y^0, y^1, y^2) = x^0 y^0 \oplus x^0 y^1 \oplus x^1 y^0.$$

That is, $f^0(x^0, x^1, x^2, y^0, y^1, y^2) \oplus f^1(x^0, x^1, x^2, y^0, y^1, y^2) \oplus f^2(x^0, x^1, x^2, y^0, y^1, y^2) = xy$ for $x = x^0 \oplus x^1 \oplus x^2$, $y = y^0 \oplus y^1 \oplus y^2$ and the computation of $f^i$ does not involve $x^i$ nor $y^i$ for each $i = 0, 1, 2$.

Finally, the shared implementation $(f^0, \ldots, f^{d_{out}})$ of $f$ is said to be uniform if, for any output mask $(z^0, z^1, \ldots, z^{d_{out}})$, the probability that $f^j(x^0, x^1, \ldots, x^{d_{in}}) = z^j$ for all $0 \leq j \leq d_{out}$, is constant if $z^0 \oplus \cdots \oplus z^{d_{out}}$ is fixed. For example, the following shared implementation of the AND gate is uniform as Table 1 shows: for random bits $r, s$ and $x^0, x^1, x^2, y^0, y^1, y^2 \in GF(2)$,

$$f^0(x^0, x^1, x^2, y^0, y^1, y^2) = z^0 = x^1 y^1 \oplus x^1 y^2 \oplus x^2 y^1 \oplus r$$

$$f^1(x^0, x^1, x^2, y^0, y^1, y^2) = z^1 = x^2 y^2 \oplus x^2 y^0 \oplus x^0 y^2 \oplus s \tag{4}$$

$$f^2(x^0, x^1, x^2, y^0, y^1, y^2) = z^2 = x^0 y^0 \oplus x^0 y^1 \oplus x^1 y^0 \oplus r \oplus s.$$

**Table 1.** Probability distribution of $(z^0, z^1, z^2)$ in (4).

| $z^0 \oplus z^1 \oplus z^2$ | | | 0 | | | | 1 | |
|---|---|---|---|---|---|---|---|---|
| $(z^0, z^1, z^2)$ | (0,0,0) | (0,1,1) | (1,0,1) | (1,1,0) | (1,0,0) | (0,1,0) | (0,0,1) | (1,1,1) |
| **Prob.** | 3/16 | 3/16 | 3/16 | 3/16 | 1/16 | 1/16 | 1/16 | 1/16 |

## 3. Non-Completeness Implies 1st-Order DPA Security?

As stated in Section 2, Threshold Implementation is known to be provably secure against the 1st-order differential power attack even in the presence of glitches, and it is usually referenced that the corresponding security proof is given in Theorem 2 and Corollary 1 of [11]. Unfortunately, as indicated below, the proof is not satisfactory in the sense that there is a counter-example of not being secure against the glitch attack while satisfying the condition presented in [11]. For easy reference, here, we re-state the theorem.

**Theorem 1** (same with Theorem 2 in [11]). *For a shared implementation $(f^0, \ldots, f^{d_{out}})$ of $z = f(x)$, if the input shares are uniform and $(f^0, \ldots, f^{d_{out}})$ is correct and non-complete, then each output share $z^i$ is statistically independent of the original input $x$ and the original output $z$. And, the same holds for any intermediate result that is appearing during the computation of $(f^0, \ldots, f^{d_{out}})$ and for any physical quantities like power consumption, electro-magnetic radiation, etc., which are a function of these intermediate results.*

Note that the original theorem in [11] was stated for the multi-input and multi-output function, while Theorem 1 above assumes that $f$ has a single input and single output. However, the difference does not affect the validity of the argument below.

In [11], for proving the theorem above, the authors showed that, for any variable $\tau$ appearing in the computation of $(f^0, \ldots, f^{d_{out}})$ and any input $x$ of $f$,

$$\Pr(\tau) = \Pr(\tau|x). \tag{5}$$

Clearly, the Equation (5) implies that $\tau$ and $x$ are statistically independent. And then, [11] presented the following corollary:

**Corollary 1** ([11]). *For a shared implementation $(f^0, \ldots, f^{d_{out}})$ of $z = f(x)$, if the input shares are uniform and $(f^0, \ldots, f^{d_{out}})$ is correct and non-complete, then the expected value of the power consumption of a circuit implementing $(f^0, \ldots, f^{d_{out}})$ is independent of $x$ and $z$, even in the presence of glitches or the delayed arrival of some inputs.*

Now, in proving the corollary above, the authors of [11] stated that "Since the proof of Theorem 2 makes no assumption on the behavior of the circuit and/or the presence of glitches, the theorem holds for each sub-circuit computing one of the $y_i^j$, also in the case of delayed inputs or glitches. Furthermore, the mean power consumption of the whole circuit is the sum of the mean power consumptions of the sub-circuits and expectation is a linear operation" [11]. After all, [11] used the argument that, if all the intermediate variables of the shared circuit are statistically independent of the input and the output, the shared implementation is secure against the 1st-order DPA even in the presence of a glitch. However, this argument is not true, in general. That is, there are some shared circuits whose intermediate variables are statistically independent of the input and the output, but that are not secure against the glitch attack. For example, consider the masking scheme by Trichina described in Figure 1. First, it can be proved that all the intermediate results of the scheme, say, $x^0y^0, x^1y^0, x^0y^1, x^1y^1, r \oplus x^0y^0, (r \oplus x^0y^0) \oplus x^1y^0,$ $((r \oplus x^0y^0) \oplus x^1y^0) \oplus x^0y^1, (((r \oplus x^0y^0) \oplus x^1y^0) \oplus x^0y^1) \oplus x^1y^1$ are statistically independent of the original input and output. More precisely, for any $\alpha, \beta \in \mathrm{GF}(2)$, if $\tau \in \left\{x^0y^0, x^1y^0, x^0y^1, x^1y^1\right\}$, then $\Pr(\tau = 0) = \Pr(\tau = 0|(x, y) = (\alpha, \beta)) = \frac{3}{4}$, while if $\tau \in \left\{r \oplus x^0y^0, (r \oplus x^0y^0) \oplus x^1y^0, ((r \oplus x^0y^0) \oplus x^1y^0) \oplus x^0y^1, (((r \oplus x^0y^0) \oplus x^1y^0) \oplus x^0y^1) \oplus x^1y^1\right\}$, then $\Pr(\tau = 0) = \Pr(\tau = 0|(x, y) = (\alpha, \beta)) = \frac{1}{2}$.

Similarly, we can prove that $\Pr(\tau = 0) = \Pr(\tau = 0|xy = \alpha\beta)$ for any $\alpha, \beta$ and $\tau \in \{x^0y^0, x^1y^0, x^0y^1,$ $x^1y^1, r \oplus x^0y^0, (r \oplus x^0y^0) \oplus x^1y^0, ((r \oplus x^0y^0) \oplus x^1y^0) \oplus x^0y^1, (((r \oplus x^0y^0) \oplus x^1y^0) \oplus x^0y^1) \oplus x^1y^1\}$. Thus, the presupposition of Corollary 1 is satisfied for the Trichina's scheme; however, the scheme is also known to be susceptible to the glitch attack [7–9], which contradicts the conclusion of Corollary 1.

**Remark 1.** *We note that the security guarantee of TI given in [11] is against the 1st-order univariate DPA. Thus, the attackers are assumed to be able to utilize only the mean value of power traces gathered at a specific time moment. And, the attackers who rely on the variance value of gathered power traces or power traces gathered at several time moments are called the higher-order DPA attackers, and [11] and this paper do not consider such attackers.*

At this point, it is emphasized that we do not claim that the non-complete TI with uniform inputs has some power attack weaknesses. Actually, the non-complete TI is shown to be very secure in various leakage detection tests [10–13]. Thus, it is plausible that TI gives a lot of resistance to DPA, even in the presence of glitches. However, as the previous argument shows, it is not clear why it gives such security from a theoretical viewpoint. After all, the problem is that [11] does not contain any theoretical explanation of how we can mitigate the glitch effect. And, as will be explained in the next section, the G-equivariance [14] gives a useful instrument for how to mitigate such an effect. However, G-equivariance also has its own drawbacks. Most notably, there are no (1st-order) G-equivariant gates, the XOR sum in which the output gives rise to the ordinary 2-input AND gate evaluation, and that is why we should use the extended concept of G-equivariance to the higher-order setting in [16]. More details can be found in Section 4.

## 4. Non-Completeness Implies G-Equivariance

To solve the problem discussed in the previous section, we use the extended concept of the G-equivariance in [16].

Originally, the G-equivariance was introduced as a glitch attack countermeasure, and a gate (or a function) is defined to be G-equivariant if the energy consumption of the gate is independent of the arrival order of input signals [14]. However, as paper [14] indicated, there are no G-equivariant gates in which the Boolean sum is equal to the 2-input AND gate evaluation, and the authors loosened the condition imposed on the G-equivariance to get the semi-G-equivariance. The semi-G-equivariance (the exact definition of which can be found in [14]) requires the energy consumption of a gate to be independent of the order of input signals with the constraint that some signals should arrive at the gate simultaneously, and [14] proved that there are some semi-G-equivariant gates that summed up to give the 2-input AND gate. For example, the gates $g^0, g^1$ with

$$g^0(x^0,\ x^1, y^0, y^1) = x^0 \oplus y^0 \tag{6}$$

$$g^1(x^0,\ x^1, y^0, y^1) = x^0 \oplus y^0 \oplus x^0 y^0 \oplus x^0 y^1 \oplus x^1 y^0 \oplus x^1 y^1$$

have the property that they are semi-G-equivariant, that is, their energy consumption is independent of the arrival order of input signals if $x^0$ and $x^1$ arrive at $g^0$ or $g^1$ at the same time and $y^0$ and $y^1$ arrive at $g^0$ or $g^1$ simultaneously, and $g^0(x^0,\ x^1, y^0, y^1) \oplus g^1(x^0,\ x^1, y^0, y^1) = (x^0 \oplus x^1)(y^0 \oplus y^1)$. However, there is a critical problem with semi-G-equivariant gates. It is not easy to satisfy the condition imposed on the gates. That is, for example, it is very hard to make $x^0$ and $x^1$ (and $y^0$ and $y^1$ as well) in (6) arrive at $g^0$ or $g^1$ at the same time. The custom design process may solve the issue; however, with a high cost.

The G-equivariance and semi-G-equivariance in [14] were basically defined in the 1st-order setting. More precisely, any input signal $x$ of a gate is assumed to be decomposed as $(x^0,\ x^1)$ with $x = x^0 \oplus x^1$. However, there is no justification for why the input signals for the G-equivariant gate should have such form. And, using the extended notion of G-equivariance to the higher-order setting in [16], this paper assumes that every signal $x$ is represented as $(x^0,\ x^1, \ldots,\ x^d)$ with $x = x^0 \oplus x^1 \oplus \cdots \oplus x^d$ for a given positive integer $d$.

In the subsequent, a gate with $n$ inputs and 1 output is considered as a Boolean function $g : \mathrm{GF}(2^n) \to \mathrm{GF}(2)$, where $\mathrm{GF}(2)$ stands for the binary finite field with addition operation $\oplus$, and $\mathrm{GF}(2^n)$ is an extension field of degree $n$ over $\mathrm{GF}(2)$, which can be considered as an $n$-dimensional vector space over $\mathrm{GF}(2)$. For a positive integer $n$, we denote $Map(n)$ as the set of all mappings from the set $\{0,\ \ldots,\ n-1\}$ into itself.

**Definition 1** ([14,16]). *Given a positive integer $n$, a gate $g : GF(2^n) \to GF(2)$ and $i = 0, 1,\ \ldots,\ n-1$, the partial energy function $E_{g,i}$ is defined as*

$$E_{g,i} : \mathrm{GF}(2^n) \times \mathrm{GF}(2^n) \times Map(n) \to V \tag{7}$$

$$((a,x),\ \phi) \oplus e_{g(b^i)g(b^{i+1})}$$

*where $V$ stands for the 4-dimensional real vector space with the basis $\{e_{00},\ e_{01}, e_{10},\ e_{11}\}$ and, for $a = (a_0,\ \ldots,\ a_{n-1})$, $x = (x_0,\ \ldots,\ x_{n-1})$ and $\phi \in Map(n)$, $b^i = (b^i_0,\ \ldots,\ b^i_{n-1}) \in GF(2^n)$ in (7) is defined by: $b^0 = a$ and*

$$b^{i+1}_j = \begin{cases} x_j & if\ \phi(j) = i \\ b^i_j & otherwise \end{cases}. \tag{8}$$

In Definition 1, the basis vectors $e_{00}, e_{01},\ e_{10}, e_{11}$ of $V$ can actually be interpreted as the amount of power consumed while $g$ changes or holds its output. That is, when the output of $g$ changes from 0 to 1, it is assumed that $g$ consumes the energy $e_{01}$, while $g$ is assumed to consume the energy $e_{10}$, if $g$

changes its output from 1 to 0. And, when the output bit of $g$ is fixed at 0 or 1, $g$ is assumed to consume the energy $e_{00}$ or $e_{11}$, respectively. Also, in the definition, $\phi \in Map(n)$ was introduced to describe the arrival order of the gate's input signal. Thus, for example, $\phi(j) = i$ in (8) implies that the $j$-th input signal $a_j$ arrives at the $i$-th order.

Since we are primarily interested in masking a 2-input AND gate, all the gates in the subsequent are assumed to have 2 inputs.

**Definition 2** ([14,16]). *For a positive integer d and any input* $(a, b) \in GF(2^2)$ *of a gate* $g : GF(2^2) \to GF(2)$, *the tuple* $(a^0, a^1, \ldots, a^d, b^0, b^1, \ldots, b^d) \in GF(2^{2(d+1)})$ *is called a d-th order masked signal of* $(a, b)$ *if*

(1)  $a = a^0 \oplus a^1 \oplus \cdots \oplus a^d$ *and* $b = b^0 \oplus b^1 \oplus \cdots \oplus b^d$
(2)  $\Pr(a^i = 1) = \Pr(b^j = 1) = \frac{1}{2}$ *for any* $i, j = 0, 1, \ldots, d$
(3)  *For any* $i, j = 0, 1, \ldots, d$, $a^i$ *and* $b^j$ *are statistically independent, considered as random variables.*

**Definition 3** ([14,16]). *A d-th order masked gate of a gate* $g : GF(2^2) \to GF(2)$ *is the tuple* $(g^0, g^1, \ldots, g^d)$ *satisfying the following:*

(1)  $g^0, g^1, \ldots, g^d : GF(2^{2(d+1)}) \to GF(2)$
(2)  *For any input* $(a, b) \in GF(2^2)$ *of g and any d-th order masked signal* $(a^0, a^1, \ldots, a^d, b^0, b^1, \ldots, b^d) \in GF(2^{2(d+1)})$ *of* $(a, b)$, *we have* $g^0(\widetilde{a}, \widetilde{b}) \oplus g^1(\widetilde{a}, \widetilde{b}) \oplus \cdots \oplus g^d(\widetilde{a}, \widetilde{b}) = g(a, b)$ *for* $(\widetilde{a}, \widetilde{b}) = (a^0, a^1, \ldots, a^d, b^0, b^1, \ldots, b^d)$.

*In this case, each* $g^k$ *of the d-th order masked gate* $(g^0, g^1, \ldots, g^d)$ *is called a d-th order component masked gate of g.*

**Definition 4** ([14,16]). *Given a d-th order masked gate* $(g^0, g^1, \ldots, g^d)$ *of a gate* $g : GF(2^2) \to GF(2)$, *a d-th order component masked gate, say* $g^k : GF(2^{2(d+1)}) \to GF(2)$ *isd-th order G-equivariant if for any* $\phi \in Map(2(d+1))$ *and* $i = 0, 1, \ldots, 2d+1$, *the expectation value* $E(E_{g^k,i}(((\widetilde{a}, \widetilde{b}), (\widetilde{x}, \widetilde{y})), \phi))$ *is independent of any choice of d-th order masked signals* $(\widetilde{a}, \widetilde{b}) = (a^0, a^1, \ldots, a^d, b^0, b^1, \ldots, b^d)$ *of* $(a, b) \in GF(2^2)$ *and* $(\widetilde{x}, \widetilde{y}) = (x^0, x^1, \ldots, x^d, y^0, y^1, \ldots, y^d) \in GF(2^2)$ *of* $(x, y) \in GF(2^2)$.

In the sequel, a *d*-th order G-equivariant component masked gate is briefly called as a G-equivariant gate, if there is no confusion.

**Lemma 1** ([14,16]). *Given a d-th order masked gate* $(g^0, g^1, \ldots, g^d)$ *of a gate* $g : GF(2^2) \to GF(2)$, *a d-th order component masked gate, say* $g^k : GF(2^{2(d+1)}) \to GF(2)$ *is d-th order G-equivariant if and only if, for any* $\phi \in Map(2(d+1))$ *and* $i = 0, 1, \ldots, 2d+1$, *the following 16 values* $S(a, b, x, y)$ *are equal for any choice* $a, b, x, y \in \{0, 1\}$:

$$S(a, b, x, y) := \sum_{\substack{a^0, a^1, \ldots, a^d, b^0, b^1, \ldots, b^d, x^0, x^1, \ldots, x^d, y^0, y^1, \ldots, y^d \\ a^0 \oplus a^1 \oplus \cdots \oplus a^d = a \\ b^0 \oplus b^1 \oplus \cdots \oplus b^d = b \\ x^0 \oplus x^1 \oplus \cdots \oplus x^d = x \\ y^0 \oplus y^1 \oplus \cdots \oplus y^d = y}} E_{g^k,i}((\widetilde{a}, \widetilde{b}), (\widetilde{x}, \widetilde{y}), \phi), \quad (9)$$

*where* $(\widetilde{a}, \widetilde{b}) = (a^0, a^1, \ldots, a^d, b^0, b^1, \ldots, b^d)$ *and* $(\widetilde{x}, \widetilde{y}) = (x^0, x^1, \ldots, x^d, y^0, y^1, \ldots, y^d)$.

Finally, we are at the moment when we can prove that non-completeness implies G-equivariance. However, since the provable security is believed to hold for TI satisfying the first-order non-completeness (in fact, the higher-order non-completeness does not guarantee the higher-order security, as shown

in [17]), and the first-order non-completeness is generally realized in the 2nd-order masking scheme ([10–13]), we will focus on the 2nd-order masked gate in the sequel.

**Theorem 2.** *Given a 2nd order masked gate* $(g^0, g^1, g^2)$ *of a gate* $g : GF(2^2) \rightarrow GF(2)$, *if a 2nd order component masked gate, say* $g^k : GF(2^6) \rightarrow GF(2)$ *is first-order non-complete, then it is 2nd-order G-equivariant.*

**Proof.** Without loss of generality, assume that $g^k(x_0, x_1, x_2, y_0, y_1, y_2)$ is independent of $x_0$ and $y_0$ so that $g^k(x_0, x_1, x_2, y_0, y_1, y_2)$ can be denoted as $g^k(x_1, x_2, y_1, y_2)$. By Lemma 1, proving that $g^k$ is 2nd-order G-equivariant is equivalent to showing that, for any $\phi \in Map(6)$ and $i = 0, 1, \ldots, 5$, $S(a, b, x, y) = S(a', b', x', y')$ for any $(a, b, x, y)$, $(a', b', x', y') \in \{0, 1\}^4$, where $S(a, b, x, y)$ is defined by

$$S(a,b,x,y) := \sum_{\substack{a^0,a^1,a^2,b^0,\ b^1,b^2,x^0,x^1,x^2,y^0,y^1,\ y^2 \\ a^0 \oplus a^1 \oplus a^2 = a \\ b^0 \oplus b^1 \oplus b^2 = b \\ x^0 \oplus x^1 \oplus x^2 = x \\ y^0 \oplus y^1 \oplus y^2 = y}} E_{g^k,i}((\widetilde{a}, \widetilde{b}), (\widetilde{x}, \widetilde{y}), \phi), \tag{10}$$

where $(\widetilde{a}, \widetilde{b}) = (a^0, a^1, a^2, b^0, b^1, b^2)$ and $(\widetilde{x}, \widetilde{y}) = (x^0, x^1, x^2, y^0, y^1, y^2)$. Note that (10) is equal to the Equation (9) for $d = 2$. Now, rewriting the summation in (10), we have

$$S(a,b,x,y) = \sum_{\substack{a^1,a^2,\ b^1,b^2,x^1,x^2,y^1,\ y^2 \\ a^0 \oplus a^1 \oplus a^2 = a \\ b^0 \oplus b^1 \oplus b^2 = b \\ x^0 \oplus x^1 \oplus x^2 = x \\ y^0 \oplus y^1 \oplus y^2 = y}} \sum_{a^0,\ b^0,x^0,\ y^0} E_{g^k,i}((\widetilde{a}, \widetilde{b}), (\widetilde{x}, \widetilde{y}), \phi). \tag{11}$$

And the Equation (11) is equal to

$$\sum_{\substack{a^1,a^2,\ b^1,b^2,x^1,x^2,y^1,\ y^2 \\ a^0 \oplus a^1 \oplus a^2 = a \\ b^0 \oplus b^1 \oplus b^2 = b \\ x^0 \oplus x^1 \oplus x^2 = x \\ y^0 \oplus y^1 \oplus y^2 = y}} \sum_{\widetilde{a}^0, \widetilde{b}^0, \widetilde{x}^0, \widetilde{y}^0} E_{g^k,i}((\widetilde{a'}, \widetilde{b'}), (\widetilde{x'}, \widetilde{y'}), \phi), \tag{12}$$

for $\Delta a = a \oplus a'$, $\Delta b = b \oplus b'$, $\Delta x = x \oplus x'$, $\Delta y = y \oplus y'$, $\widetilde{a}^0 = a^0 \oplus \Delta a$, $\widetilde{b}^0 = b^0 \oplus \Delta b$, $\widetilde{x}^0 = x^0 \oplus \Delta x$, $\widetilde{y}^0 = y^0 \oplus \Delta y$, $(\widetilde{a'}, \widetilde{b'}) = (\widetilde{a}^0, a^1, a^2, \widetilde{b}^0, b^1, b^2)$ and $(\widetilde{x'}, \widetilde{y'}) = (\widetilde{x}^0, x^1, x^2, \widetilde{y}^0, y^1, y^2)$ since $g^k(x_0, x_1, x_2, y_0, y_1, y_2)$ is independent of $x_0$ and $y_0$, thus changing $a^0$ and $b^0$ to $\widetilde{a}^0$ and $\widetilde{b}^0$ in $(\widetilde{a}, \widetilde{b}) = (a^0, a^1, a^2, b^0, b^1, b^2)$ and changing $x^0$ and $y^0$ to $\widetilde{x}^0$ and $\widetilde{y}^0$ in $(\widetilde{x}, \widetilde{y}) = (x^0, x^1, x^2, y^0, y^1, y^2)$ does not impact on evaluating $g^k$ and $E_{g^k,i}$. Finally, substituting $a^0$ with $\widetilde{a}^0 \oplus \Delta a$, $b^0$ with $\widetilde{b}^0 \oplus \Delta b$, $x^0$ with $\widetilde{x}^0 \oplus \Delta x$ and $y^0$ with $\widetilde{y}^0 \oplus \Delta y$, the Equation (12) is equal to

$$\sum_{\substack{a^1,a^2,\ b^1,b^2,x^1,x^2,y^1,\ y^2 \\ \widetilde{a}^0 \oplus a^1 \oplus a^2 = a' \\ \widetilde{b}^0 \oplus b^1 \oplus b^2 = b' \\ \widetilde{x}^0 \oplus x^1 \oplus x^2 = x' \\ \widetilde{x}^0 \oplus x^1 \oplus x^2 = x'}} \sum_{\widetilde{a}^0, \widetilde{b}^0, \widetilde{x}^0, \widetilde{y}^0} E_{g^k,i}((\widetilde{a'}, \widetilde{b'}), (\widetilde{x'}, \widetilde{y'}), \phi), \tag{13}$$

which is again equal to

$$S(a', b', x', y') = \sum_{\substack{\widetilde{a}^0, a^1, a^2, \widetilde{b}^0, b^1, b^2, \widetilde{x}^0, x^1, x^2, \widetilde{y}^0, y^1, y^2 \\ \widetilde{a}^0 \oplus a^1 \oplus a^2 = a' \\ \widetilde{b}^0 \oplus b^1 \oplus b^2 = b' \\ \widetilde{x}^0 \oplus x^1 \oplus x^2 = x' \\ \widetilde{y}^0 \oplus y^1 \oplus y^2 = y'}} E_{g^k, i}((\widetilde{a'}, \widetilde{b'}), (\widetilde{x'}, \widetilde{y'}), \phi) \qquad (14)$$

And, this completes the proof. □

**Corollary 2.** *The following 2nd-order component masked gates* $f^0, f^1, f^2 : GF(2^6) \to GF(2)$ *give the shared implementation* $(f^0, f^1, f^2)$ *of the 2-input AND gate whose energy consumption is independent of the arrival order of input signals:*

$$f^0(x^0, x^1, x^2, y^0, y^1, y^2) = x^1 y^1 \oplus x^1 y^2 \oplus x^2 y^1$$
$$f^1(x^0, x^1, x^2, y^0, y^1, y^2) = x^2 y^2 \oplus x^2 y^0 \oplus x^0 y^2 \qquad (15)$$
$$f^2(x^0, x^1, x^2, y^0, y^1, y^2) = x^0 y^0 \oplus x^0 y^1 \oplus x^1 y^0.$$

*In other words, the energy consumption of* $f^0, f^1, f^2$ *in (15) is independent of the arrival order of input signals and* $f^0(x^0, x^1, x^2, y^0, y^1, y^2) \oplus f^1(x^0, x^1, x^2, y^0, y^1, y^2) \oplus f^2(x^0, x^1, x^2, y^0, y^1, y^2) = (x^0 \oplus x^1 \oplus x^2)(y^0 \oplus y^1 \oplus y^2).$

**Proof.** The claim is a direct consequence of Theorem 2 since $f^0, f^1, f^2$ are first-order non-complete. □

By Corollary 2, the set of first-order non-complete gates is a subset of the set of 2nd-order G-equivariant gates. And, at this point, it may be questionable if there are any 2nd-order G-equivariant but first-order complete gates that can be used for implementing the AND gate in a shared form. Interestingly, the answer is yes. For example, it can be shown that the following gates $g^0, g^1, g^2$ are G-equivariant:

$$g^0(x^0, x^1, x^2, y^0, y^1, y^2) = x^0 y^0 \oplus x^0 y^1 \oplus x^1 y^0 \oplus x^1 y^2 \oplus x^2 y^1 \oplus x^2 y^2$$

$$g^1(x^0, x^1, x^2, y^0, y^1, y^2) = x^0 y^2 \oplus x^2 y^0 \oplus x^2 y^2 \qquad (16)$$

$$g^2(x^0, x^1, x^2, y^0, y^1, y^2) = x^1 y^1 \oplus x^2 y^2.$$

Also, the XOR sum of $g^0, g^1, g^2$ in (16) gives the AND gate evaluation for the original inputs, that is, $g^0(x^0, x^1, x^2, y^0, y^1, y^2) \oplus g^1(x^0, x^1, x^2, y^0, y^1, y^2) \oplus g^2(x^0, x^1, x^2, y^0, y^1, y^2)$ is equal to $(x^0 \oplus x^1 \oplus x^2)(y^0 \oplus y^1 \oplus y^2)$. However, since $g^0$ involves all shares $x^0, x^1, x^2$ of $x$ for its computation, it is not non-complete.

**Remark 2.** *The G-equivariant gate is for ensuring the security only for the single gate; thus, it does not guarantee the security of the composition of several gates. And, to get the security of composited gates, the cryptosystem's implementers must consider inserting some registers, for example, to eliminate the glitch's effect to not propagate through the several gates. However, we emphasize that the same undesirable increase of circuit's size from inserting registers is very common in most hardware masking schemes, including the threshold implementation [10–13], mainly due to the glitch's effect.*

## 5. Conclusions

In this paper, we re-investigated the proof that TI is secure against the 1st-order DPA even in the presence of glitches and argued that the proof is missing some points. To remedy it, we proposed to

utilize the extended concept of G-equivariance to a higher-order setting. Also, this paper proves that any non-complete masked gates are actually G-equivariant, which implies that any non-complete TI can successfully prevent the glitch attack. Finally, we show that there are some G-equivariant gates that are complete; thus, the notion of G-equivariance is broader than that of non-completeness.

**Funding:** This paper is a result that was implemented as a research project on Efficiency and Security of Higher-Order Threshold Implementation by the affiliated institute of ETRI. When giving a presentation on this report, the presenter has to clarify that it is the research by the affiliated institute of ETRI.

**Conflicts of Interest:** The author declares no conflict of interest.

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
