# Peer review of "On Non-Completeness and G-Equivariance"

_applsci, doi:10.3390/app9214692_

Round 1

Reviewer 1 Report

This paper discusses a theoretical aspect of a countermeasure against the side-channel attack. There are two major problems.

First, since the definitions (Definition 4-4 etc.) and notations are ambiguous and confusing, it is hard to check the statement of theorems (Theorem 4-1, etc.).

Second, this paper seems to discuss the security only for the single gate (e.g., Theorem 4-1). In this paper, glitches within a composite circuit seem not captured. Of course, if evaluated value for each gate is hold with a register (namely, the circuit evaluate one gate per one clock cycle), the security of the circuit can be discussed as this paper claims; however, such circuit requires huge numbers of clock cycles and inefficient.

Moreover, some notations are misleading. For example, at line 281, x_d means the d-th share of x; on the other hand, at line 308, x_d means the d-th bit of x. And, at line 312, i and j are bit {0,1}; on the other hand, at line 314, i and j are integer in [0, n-1]. The authors should use the different notations for the different usage.

In Definition 4-1, there is no explanation of \phi. The authors should explain when and how to choose \phi.

Definition 4-4 seems strange. At first, at line 334, the inputs of E_{g,i} seem wrong. Its first and second input should be n-bit string; however, at the line, \tilde{a} and \tilde{b} are (d+1) n-bit strings each. Moreover, although the definition of original G-equivalence is given for the arrival order of signals, the definition here is given for the masked signal. Assume the circuit where the signals are divided in d-shares and the means of E_{g,i} is independent of choice of the mask (share), but the variance depends on the choice. Such circuit satisfies the condition of the definition; however, it might be insecure against the glitch attack.

At line 342, four elements (\tilde{a},\tilde{b},\tilde{x},\tilde{y}) are input the E_{g,i}, although E_{g,i} has only two inputs. It seems strange.

Editorial comments:

Line 255: Corollary 1 -> Corollary 3-1

Line 310: \varphi -> \phi

Line 320 (and more): F_2^{d+1} -> F_2^{2(d+1)}

Line 322: Pr(b_j) -> Pr(b_j=1)

Line 342: Bottom two lines below \Sum are not conditions but the notations. They should be written after the equation such as, \[S(a,b,x,y) … E_{g,i}(…), \] where (\tilde{a},\tilde{b})=…

Author Response

Thank you for your valuable comments.

Please, see the attached file for my response to your comments. 

Reviewer 2 Report

See the attached review in PDF.

Author Response

(The authors gave the same response as above.)
